# Feeding Calcium-Ammonium Nitrate to Lactating Dairy Goats: Milk Quality and Ruminal Fermentation Responses

**DOI:** 10.3390/ani12080983

**Published:** 2022-04-11

**Authors:** Kleves V. Almeida, Geraldo T. Santos, Jesus A. C. Osorio, Jean C. S. Lourenço, Monique Figueiredo, Thomer Durman, Francilaine E. Marchi, Claudete R. Alcalde, Ranulfo C. Silva-Junior, Camila C. B. F. Itavo, Rafael C. Araujo, Andre F. Brito

**Affiliations:** 1Department of Animal Science, State University of Maringa, Maringa 87020, Brazil; kleves.almeida@unh.edu (K.V.A.); jaco.mvz@hotmail.es (J.A.C.O.); jeancarloslsss@gmail.com (J.C.S.L.); figueiredomonique@hotmail.com (M.F.); thomerdurman@hotmail.com (T.D.); francieloise@hotmail.com (F.E.M.); cralcalde@wnet.com.br (C.R.A.); 2Department of Agriculture, Nutrition, and Food Systems, University of New Hampshire, Durham, NH 03824, USA; andre.brito@unh.edu; 3Department of Chemistry, State University of Maringa, Maringa 87020, Brazil; rcsjunior@uem.br; 4Faculty of Veterinary Medicine and Animal Science, Federal University of Mato Grosso do Sul, Campo Grande 79070, Brazil; camila.itavo@ufms.br; 5GRASP Ind. & Com. LTDA, Curitiba 81260, Brazil; rafael@grasp.ind.br

**Keywords:** antioxidant capacity, feed additive, hydrogen sink, nitrite, non-protein nitrogen

## Abstract

**Simple Summary:**

Calcium-ammonium nitrate (CAN) has been extensively used as a potential methane inhibitor for ruminants; however, there is still a need for studies focused on investigating its effects on the fatty acid profile and antioxidant capacity of milk, especially from dairy goats. Therefore, we evaluated the effects of CAN on nutrient digestibility, ruminal fermentation, and milk quality of lactating Saanen goats. Treatments consisted of a control diet (without CAN), 10 g of CAN per kg of dry matter, and 20 g of CAN per kg of dry matter. Supplemental CAN did not affect feed intake, digestibility of nutrients, and most ruminal fermentation parameters. Yields and composition of milk were not affected, and minor treatment effects were observed on the milk fatty acid profile. Milk antioxidant capacity was altered by increased conjugated dienes and reduced thiobarbituric acid reactive substances, along with greater concentrations of nitrate and nitrite residues in milk. Calcium-ammonium nitrate can be fed to lactating dairy goats up to 20 g per kg of dry matter without negative effects on nutrient digestibility and milk composition; however, it increased the concentration of conjugated dienes in milk, which may induce its faster lipid oxidation.

**Abstract:**

We aimed to investigate the effects of calcium-ammonium nitrate (CAN) fed to lactating dairy goats on dry matter (DM) intake, digestibility of nutrients, milk properties (composition, antioxidant capacity, fatty acid profile, and nitrate residues), and ruminal fermentation parameters. Twelve lactating Saanen goats averaging 98.5 ± 13.1 days in milk, 53.5 ± 3.3 kg of body weight, and 2.53 ± 0.34 kg of milk/day were randomly assigned in four 3 × 3 Latin squares to receive the following diets: a control group (without CAN) with 7.3 g/kg DM of urea (URE), 10 g/kg DM of CAN (CAN10), and 20 g/kg DM of CAN (CAN20). Each period lasted 21 days, with 14 days for diet adaptation and seven days for data and sample collection. The DM intake, digestibility of nutrients, yields of milk, 3.5% fat-corrected milk, and energy-corrected milk were not affected by treatments. Similarly, there were no treatment effects on the yields and concentrations of milk fat, true protein, and lactose, along with minor effects on milk fatty acid profile. Total antioxidant capacity in milk was unaffected by treatments; however, concentration of conjugated dienes increased, while thiobarbituric acid reactive substances in milk decreased linearly. Nitrate and nitrite residues in milk were elevated by treatments, while the total of volatile fatty acids and ammonia-N concentration in the rumen were unaffected. Collectively, feeding CAN (up to 20 g/kg of DM) to lactating dairy goats did not affect feed intake, nutrient digestibility, and milk composition; however, it may increase milk lipid oxidation, as evidenced by increased conjugated diene concentration.

## 1. Introduction

Nitrate (NO_3_^−^) is an inorganic anion that has been largely used in ruminant diets as a potential feed additive to inhibit enteric methane (CH_4_) production and concomitantly as a source of non-protein nitrogen (NNP) due to its capacity to provide ammonia (NH_3_) to ruminal microbes [1,2]. Methane suppression within the rumen occurs because NO_3_^−^ is an electron acceptor that acts at the expense of methanogenesis as metabolic hydrogen [H] sink, and because of the reduction from NO_3_^−^ to NH_3_ via denitrification, it may generate intermediates such as nitrous oxide (N_2_O), nitric oxide (NO), and nitrite (NO_2_^−^), the latter considered toxic for ruminal methanogens [3,4,5,6].

It is well-established that supplementing NO_3_^−^ to ruminants must be conducted cautiously and an adaptation period with incremental doses of NO_3_^−^ in the diet is highly recommended to avoid animal poisoning due to an unwanted NO_2_^−^ accumulation in the rumen, which may also be absorbed through the rumen wall and transferred to tissues and animal products such as milk [7,8,9]. Indeed, previous studies showed that feeding NO_3_^−^ to dairy cows with a prior adaptation period warranted low NO_3_^−^ and NO_2_^−^ residues in milk and therefore without risks for food safety [10,11,12]. Nevertheless, additional studies investigating residues of NO_3_^−^ and NO_2_^−^ in milk from treated dairy goats are required, mainly due to its importance for human nutrition, considering singularities such as lower allergenic properties when compared to the milk from dairy cows [13].

In addition, there is still limited information behind the mechanism of NO_3_^−^ and NO_2_^−^ transference to milk, and how these residues would affect milk antioxidant capacity. As mentioned previously, NO can also be produced during the reduction of NO_3_^−^ to NH_3_. Thus, it is speculated that the presence of NO might may affect the milk antioxidant capacity because of its potential to induce nitrosative stress and impact milk oxidative stability [14,15]. To our knowledge, the effects of NO_3_^−^ supplementation to lactating dairy goats on milk properties such as fatty acid (FA) profile and antioxidant capacity have never been reported in the literature.

Despite the well-documented effect of NO_3_^−^ supplementation at mitigating enteric CH_4_ production [1,8] and responses such as reduced dry matter intake (DMI) caused by aspects such as lower diet palatability [12,16], there is still a discrepancy between studies on how nitrate would affect ruminal fermentation, with exception to a greater acetate proportion response [17,18]. More studies are warranted to clarify NO_3_^−^ responses on nutrient digestibility and potential changes in ruminal fermentation, which may affect the synthesis of milk components and FA profile.

We hypothesized that feeding calcium-ammonium nitrate (CAN) gradually to lactating dairy goats would not affect animal performance and nutrient digestibility, increasing acetate proportion, although without negative impacts on milk quality due to low transference of NO_3_^−^ and NO_2_^−^ residues to milk, regardless of the dose supplemented. Therefore, the objectives were to investigate the effects of incremental amounts of CAN fed to lactating dairy goats on dry matter intake, nutrient digestibility, milk production and composition, milk FA profile and antioxidant capacity, and ruminal fermentation parameters.

## 2. Materials and Methods

Experimental procedures involving animals were approved by the Animal Care Ethics Committee of the State University of Maringa to meet the guidelines of the National Council for the Control of Animal Experimentation (protocol 9512221018). The experiment was conducted at the goat unit of the State University of Maringa, Maringa, Parana, Brazil.

### 2.1. Animals, Experimental Design, and Diets

Twelve lactating Saanen goats averaging 98.5 ± 13.12 days in milk, 53.5 ± 3.34 kg of body weight (BW), and 2.53 ± 0.342 kg of milk/day (mean ± SD) were distributed in four 3 × 3 Latin squares balanced for carryover effects. The experiment lasted 63 days, divided into three periods of 21 days, with 14 days for adaptation to the experimental diets and seven days for sampling and data collection. Animals were housed in pens and fed individually to assess the DMI. Experimental diets were formulated to meet the NRC [19] requirements for lactating goats weighing 60 kg and producing 3 kg of milk/day (Table 1).

Feed ingredients (corn silage, ground corn, and soybean meal) were analyzed for chemical composition prior to the formulation of diets and corn silage was sampled weekly throughout the study and analyzed for dry matter (DM) to maintain the same forage-to-concentrate ratio. The source of NO_3_^−^ used in the study was the calcium-ammonium nitrate decahydrate [5Ca(NO_3_)·2NH_4_NO_3_·10H_2_O], with 85.0% of DM, 16.5% of N, 19.6% of Ca, and 76.5% NO_3_^−^ on a DM basis (Yara, Oslo, Norway).

Experimental diets were URE: 7.32 g of urea/kg of DM as a control group, CAN10: 10 g of CAN (7.65 g/kg of NO_3_^−^ on a DM basis), and CAN20: 20 g of CAN (15.3 g/kg of NO_3_^−^ on a DM basis). Animals were pre-adapted to the treatments during the first four days of each experimental period, with CAN added gradually (increasing 25% per day) until reaching the amount established for each treatment. Experimental diets were fed as total mixed ration (TMR) twice per day at 0800 and 1600 h in proportions of 70 and 30% of the total DMI, respectively. Diets were adjusted daily to allow approximately 5% of refusals and animal BW was recorded at the end of each experimental period before the morning feeding.

### 2.2. Sample Collection and Chemical Analyses

Data and sample collections were performed in the last seven days of each experimental period. Fecal grab samples (~30 g) were collected once daily from days 15 to 21 at different time points (0600, 0800, 1000, 1200, 1400, 1600, and 1800 h on days 15, 16, 17, 18, 19, 20, and 21, respectively) and frozen at −20 °C until analyses. Samples of concentrate, corn silage, and refusals were collected from days 15 to 20 and frozen at −20 °C for later chemical composition analyses. Feed, refusal, and fecal samples were dried at 60 °C for 48 h in a forced-air oven (Heratherm OMS180; Thermo Fisher Scientific, Waltham, MA, USA) to determine DM concentration. Samples were ground to pass through a 4-mm sieve and then to a 1-mm sieve using a Wiley mill (Thomas Scientific, Swedesboro, NJ, USA) before chemical analyses.

Concentrate and corn silage were pooled individually to obtain one sample per period, while fecal and refusal samples were composited proportionally based on their DM concentration to yield one sample per animal per period. All samples were analyzed according to AOAC [20] for DM (method 934.01), crude protein (CP; method 990.03), neutral detergent fiber (NDF; method 2002.04), ash (method 942.05), and ether-extract (EE; method 920.39). Fecal excretion was estimated according to the methodology proposed by Cochran et al. [21]. In brief, approximately 0.5 g of feed, feces, and refusals were weighed into Ankom F57 bags (25-µm porosity; Ankom Technology, Macedon, NY, USA) and incubated in the rumen of two Holstein ruminally cannulated cows. Animals were housed in a tie-stall barn with free access to water and individually fed a diet composed of 60% of corn silage and 40% of grain mix (on a DM basis). The ruminal incubations lasted 288 h and cannulas were checked twice daily to guarantee the welfare of the animals. After removing from the rumen, bags were drained, rinsed, and then analyzed for NDF in a Ankom 200 Fiber Analyzer (Ankom Technology, Macedon, NY, USA).

### 2.3. Milk Collection and Laboratory Assays

Goats were manually milked twice daily (0800 and 1600 h) and milk yield was weighed and recorded during the last seven days of each experimental period. Milk samples were collected on days 15 and 16 during each milking (morning and afternoon) and mixed proportionally according to the milk yield. A 50 mL aliquot of milk was collected and preserved with 2-bromo-2-nitropopano-1.3-diol for analyses of fat (%), protein (%), and lactose (%) by mid infrared spectrophotometry (Bentley 2000; Bentley Instrument Inc., Chaska, MN, USA) according to De Andrade [22], and milk urea nitrogen (MUN) using the Berthelot method (Chemspec 150; Bentley Instrument Inc., Chaska, MN, USA). Somatic cell count (SCC) was conducted using the Somacount FC (Bentley Instrument Inc., Chaska, MN, USA) based on Arcuri [23]. Yields of 3.5% fat corrected milk (FCM) and energy corrected milk (ECM) were calculated according to Sklan et al. [24] and Sjaunja et al. [25], respectively. Feed efficiency (FE) was calculated by the ratio between ECM yield and DMI. Five additional aliquots of milk (50 mL each) including backup samples were collected and frozen at −20 °C for later analyses of antioxidant capacity, NO_3_^−^ and NO_2_^−^ residues, and FA profile.

Conjugated dienes (CD) were measured at 232 nm by a UV–Vis spectrophotometer (Spectrum SP-2000, Castelnuovo, DB, Italy) and the results were expressed as mmol/kg of fat [26]. Thiobarbituric acid reactive substances (TBARS) were analyzed according to Vyncke [27] with modifications [28] using a UV–Vis spectrophotometer (Spectrum SP-2000, Castelnuovo, DB, Italy) with readings at 532 nm and results were expressed as mmol of malonaldehyde/kg of fat. Total antioxidant capacity (TAC) was determined as described by Nenadis et al. [29], with readings at 734 nm using a UV–Vis spectrophotometer (Spectrum SP-2000, Castelnuovo, DB, Italy) and results were expressed in Trolox equivalent (μM Trolox/mL).

Concentration of NO_3_^−^ residues in milk was obtained by alkaline catalytic oxidation, which converts nitrogenous compounds into NO_3_^−^. Subsequently, NO_3_^−^ was reduced to NO_2_^−^ using the cadmium metal and determined by diazotization with sulfanilamide and N-naphthyl (1-naphthyl-ethylenediamine-dihydrochloride) according to Cortas and Wakid [30].

Milk FA profile was analyzed via fat extraction, first by centrifugation as proposed by Murphy et al. [31] and then by esterification according to the ISO 5509 method [32] using KOH/methanol and n-heptane. Fatty acid methyl esters were quantified using a gas chromatograph (Trace GC 52 Ultra; Thermo Scientific, Waltham, MA, USA) equipped with a flame ionization detector at 240 °C and a fused silica capillary column (100 m in length, 0.25 mm internal diameter, and 0.20 μm; Restek 2560, Thermo Scientific, Waltham, MA, USA). Gas flow rate was 45 mL/min for H_2_ (carrier gas), 45 mL/min for N_2_ (auxiliary gas), and 45 to 400 mL/min of synthetic air (flame gas). Column temperature was initially set at 50 °C (4 min), raised gradually (10 °C/min) up to 200 °C (15 min), and finally raised (20 °C/min) to reach 240 °C (8 min) as the final temperature. Milk FAs were quantified by comparing the retention time of FA methyl esters from standards (18919-1 Sigma Aldrich, St. Louis, MO, USA) and milk samples.

### 2.4. Blood and Ruminal Fluid Collections

Blood was sampled by puncture of the jugular vein on day 19 of each experimental period before (0 h) and 4 h after the morning feeding into serum separator evacuated tubes, centrifuged (3200× *g* for 15 min at 4 °C), and stored at −20 °C for subsequent analyses. Concentration of plasma urea nitrogen (PUN) was analyzed colorimetrically by commercial kits (Gold Analisa, Belo Horizonte, MG, Brazil) using a spectrophotometer (Bio-2000; Bioplus, São Paulo, SP, Brazil).

Ruminal fluid was collected on day 20 of each experimental period using an esophageal tube coupled to a vacuum pump about 2 and 8 h after morning feeding. An aliquot of 50 mL was collected, and pH was immediately measured using a pH meter (Tecnal, Piracicaba, SP, Brazil). A second aliquot of 50 mL was filtered through four layers of cheesecloth, acidified with 1 mL of sulfuric acid (1:1 vol/vol), and stored at −20 °C for later analyses. Concentration of volatile fatty acids (VFAs) was determined using a gas chromatograph (Shimadzu GC-2010 Plus; Shimadzu, Kyoto, Japan) equipped with an AOC-20i automatic injector, Stabilwax-DA capillary column (30 m, 0.25 mm ID, 0.25 μm; Restek, Bellefonte, PA, USA) and a flame ionization detector after acidifying with 1 M of phosphoric acid and spiked with a water-soluble FA-2 standard. A 1 μL aliquot of each sample was injected with a 40:1 split rate using H_2_ as the carrier gas. Injector and detector temperatures were 250 °C and 300 °C, respectively. Column temperature started at 40 °C, was raised to 120 °C at a rate of 40 °C/min, followed by a gradient of 120 °C to 180 °C at the rate of 10 °C/min and a rate of 120 °C/min for 180 °C to 240 °C, and then the temperature was maintained at 240 °C for an additional 3 min. Ammonia-N (NH_3_-N) concentration was measured via colorimetric quantification of N using the phenol-hypochlorite reaction according to Broderick and Kang [33].

### 2.5. Statistical Analyses

Data were checked for the normality of residuals using the Shapiro–Wilk test. Responses that violated the assumptions of normality (ruminal NH_3_–N) were subjected to power transformation as described by Box and Cox [34]. The LSM and SEM were back transformed prior to the presentation of results [35].

The MIXED procedure of SAS (SAS ver. 9.4, SAS Institute Inc., Cary, NC, USA) was used to analyze the data according to the following model:
Y*_ijkl_* = *μ* + S*_i_* + P*_j_* + A(S)*_ki_* + T*_l_* + є*_ijkl_*,(1)
where Y*_ijkl_* = dependent variable; *μ* = overall mean; S*_i_* = random effect of *i*-th square (*i* = 1 to 4); P*_j_* = random effect of the *j*-th period (*j* = 1 to 3); A(S)*_ki_* = random effect of the *k*-th animal nested within the *i*-th square; T*_l_* = fixed effect of the *l*-th treatment (1 = URE, 2 = CAN10, and 3 = CAN20); and є*_ijkl_* = residual error associated with each observation as a random effect.

Orthogonal polynomial contrasts were used to determine linear and quadratic effects of treatments on the responses analyzed. Treatment significances and trends were declared at *p* ≤ 0.05, and 0.05 < *p* ≤ 0.10, respectively.

## 3. Results

### 3.1. Dry Matter Intake and Nutrient Digestibility

Supplemental CAN did not affect BW and DMI, averaging 53.2 kg and 1.77 kg/day, respectively (Table 2). Apparent digestibility of DM (*p* = 0.08), organic matter (OM; *p* = 0.09), and CP (*p* = 0.06) tended to a quadratic response by supplementing CAN to lactating dairy goats. However, treatment did not affect EE and NDF digestibility (Table 2).

### 3.2. Yield, Composition, Antioxidant Capacity, and Nitrate and Nitrite Residues in Milk

Feeding CAN to lactating dairy goats did not affect the yields of milk, 3.5% FCM, and ECM, with means of 2.10, 2.03, and 1.97 kg/day, respectively (Table 3). Likewise, FE was similar between treatments. Additionally, treatment had no effect on the concentrations (%) or yields (kg/d) of fat, true protein, and lactose (Table 3).

Treatment did not affect the TAC of milk (average = 203.1 μM of Trolox equivalent/mL), whereas TBARS concentration linearly reduced (*p* < 0.01) as the levels of CAN increased. In contrast, the concentration of CD in milk linearly increased (*p* = 0.02) according to the increment of CAN in the diet, while no treatment effect was observed on MUN concentration. Somatic cell count and Log_10_ SCC were not affected by treatment (Table 3).

Concentration of NO_3_^−^ residue in milk linearly increased (*p* < 0.01; URE = 0.33 mg/L vs. CAN10 = 0.31 mg/L vs. CAN20 = 0.44 mg/L; Figure 1a) with CAN supplementation, while a quadratic response (*p* = 0.03) was observed on the concentration of NO_2_^−^ in milk, with the maximum concentration for CAN10 (0.065 mg/L), followed by CAN20 (0.056 mg/L), and URE (0.042 mg/L), as shown in Figure 1b.

Dietary CAN did not affect the proportions of most saturated FA (SFA; 6:0, 8:0, 10:0, 13:0, 14:0, 15:0, 16:0, 18:0, 20:0) and monounsaturated FA (MUFA; 14:1, 15:1, 16:1, 17:1; Table 4). Additionally, there were no treatment effects on individual polyunsaturated FA (PUFA; *trans*-6 18:2, *cis*-6 18:2, *cis*-9-*trans*-11 CLA), while linear trends were observed for FA proportions of 11:0 (*p* = 0.08) and 17:0 (*p* = 0.09). The milk FA proportion of 12:0 was reduced (*p* = 0.04; linear effect), whereas *trans*-9 18:1 (*p* = 0.03) linearly increased. Total SFA, MUFA, and PUFA were not affected by CAN supplementation (Table 4).

Concentration of PUN presented a quadratic response (*p* = 0.02) by supplemental CAN and a linear increase (*p* < 0.01) over time, with the greatest levels observed after 4 h of the morning feeding; however, no interactions were observed between CAN versus time (Figure 2).

There were no interactions (data not shown) between the treatment and collection time (2 and 8 h after the morning feeding) for all ruminal fermentation parameters. Therefore, time of collection was not considered for the final presentation of results.

Supplemental CAN did not affect the ruminal pH, NH_3_-N, and Total VFA, with means of 7.2, 15.3 mM, and 52.3 mM, respectively (Table 5). Similarly, there were no treatment effects on the proportions of acetate, isobutyrate, butyrate, and isovalerate, whereas propionate proportion linearly decreased (*p* = 0.01) and acetate:propionate ratio linearly increased (*p* < 0.01; Table 5).

## 4. Discussion

### 4.1. Feed Intake and Nutrient Digestibility

We evaluated the supplementation of CAN (up to 20 g/kg on a DM basis) to lactating dairy goats and observed no treatment effects on DMI. According to Lee and Beauchemin [7], nitrate has a bitter taste, which might negatively affect feed intake in ruminants. Such effects were previously observed by De Raphélis-Soissan et al. [18] in sheep by supplementing 31 g of calcium-ammonium nitrate (~20 g/kg of NO_3_^−^ on DM basis) compared to urea. In contrast, corroborating our findings, others have observed no effects on DMI of dairy cows by supplementing CAN up to 27.9 g/kg of DM [11] or when feeding sodium nitrate (14.6 g on DM basis) as a urea replacer in low protein diets [2]. In the present study, the treatment was gradually included in the diet during the adaptation period and provided as a TMR to avoid sorting, which could partially explain the absence of effects on DMI. In addition, another reason is that the availability of NO_3_^−^ toward the rumen in both treatments (CAN10 = 7.65 g of NO_3_^−^ and CAN20 = 15.30 g of NO_3_^−^) was relatively lower when compared to studies with small ruminants that observed the effects of NO_3_^−^ on DMI [18,36].

Supplemental CAN caused only minor effects on nutrient digestibility, evidenced by quadratic trends on DM, OM, and CP. A previous in vitro study of Zhou et al. [37] showed that higher levels of supplemental NO_3_^−^ (~48 g/kg on a DM basis) reduced cellulolytic bacteria population, which may cause negative effects on NDF digestibility. We believe that the level of NO_3_^−^ used in our study (up to 15.30 g of NO_3_^−^ on DM basis), included gradually in the diet during an adaptation period of 14 days, was appropriate to avoid negative effects on ruminal microbiota, validated by the absence of treatment effects on DMI. Corroborating our findings, others observed no effects of supplemental CAN (up to 27.9 g/kg of DM) to lactating dairy cows on DM, OM, CP, and NDF digestibility in the rumen, small intestine, and hindgut [11]. In addition, Wang et al. [2], supplemented sodium nitrate (14.6 g of NO_3_^−^ on a DM basis) to dairy cows and observed no negative effects on fiber digestibility, supported by the absence of changes in cellulolytic bacteria (*Ruminococcus albus*, *R. flavefaciens,* and *Fibrobacter succinogenes*).

### 4.2. Milk Quality and Nitrate and Nitrite Residues in Milk

Dietary CAN did not affect any milk performance parameters (milk yield, 3.5% FCM, ECM, or FE). This absence of treatment response is highly related to the also unchanged milk composition in the present study. We can assume that the lack of effects on DMI, nutrient digestibility, and some VFA concentrations (acetate) can be attributable to the non-significant effect observed for milk fat. In addition, our study did not observe the effects on yield and the composition of milk true protein, which may be related as a response to the unchanged DMI. Indeed, previous studies have shown that lower DMI in response to NO_3_^−^ supplementation may cause negative effects on milk performance because the lower ingestion of nutrients can lead to a lack of gluconeogenic precursors, and consequently, affect the synthesis of milk components [10,12,38]. In line with our findings, Olijhoek et al. [11] demonstrated that supplemental CAN (up to 21.1 g/kg on a DM basis) fed to dairy cows did not change milk yield or ECM, validated by the unchanged DMI.

To our knowledge, the antioxidant capacity of milk from dairy goats supplemented with CAN has not been reported yet in the literature. Overall, TAC was not affected by treatments; however, the concentration of TBARS decreased and CD increased. Conjugated dienes are considered indicators of lipid oxidation, which may influence milk antioxidant capacity as a response to reduced-fat stability [39]. There are many methods (e.g., physical and chemical) for evaluating the oxidative stability of fats. Each method provides information about a particular state of the oxidative process, which depends on the conditions applied and the lipid substrates evaluated. The oxidation of PUFA results in the formation of hydroperoxides and displacement of double bonds, with consequent formation of CD, corresponding to the primary product from the oxidative process [40,41]. The increase in the concentration of CD was also observed earlier [42] by supplementing CAN (up to 30 g/kg of DM) to lactating dairy cows, which seemed to be associated with an initial oxidation process of the milk fat, supported by trends to increase MUFA and PUFA in milk from treated cows [42]. Thus, the supply of CAN, in a way, may have anticipated the beginning of the milk oxidation process. A possible explanation for this effect is that there was increased production of NO in milk as a product from the NO_3_^−^ to NH_3_ conversion. It is believed that increased NO, along with NO_2_^−^, can induce the nitrosative stress of milk fat by accumulating lipid peroxides, and consequently alter the milk antioxidant capacity [43]. Despite the unchanged milk SCC in the present study, milk oxidative stress can also be associated with subclinical mastitis, as observed by Silanikove et al. [43], who also suggested that milk from dairy goats was less susceptible to nitrosative stress than milk from dairy cows because of its increased TAC.

During the lipid peroxidation, compounds such as aldehydes can be formed as secondary products [44]. The nature and relative proportions of aldehydes from degradation processes are highly dependent on the type of FA oxidized and the oxidation conditions. For this evaluation, the most widely used test is the TBARS, which is based on the reaction of thiobarbituric acid with the decomposition products of hydroperoxides. One of the main products formed in the oxidative process is malondialdehyde [41]. The reduction in TBARS concentration with NO_3_^−^ supplementation in our study may be associated with a positive effect, indicating greater resistance to secondary spoilages and therefore increasing the shelf life of milk products [26]. Additionally, the absence of effects on milk TAC in the present study can also be associated with a positive effect, as this parameter generally provides information about the status of the antioxidant potential of milk, which was previously reported to be higher in dairy goats when compared to the milk from dairy cows [45].

Nitrate residue in milk was higher when the levels of CAN increased, although the maximum concentration observed (0.44 mg/L for 20 g of CAN) was still under the recommendations of the WHO [46] for human consumption, which is limited to 50 mg of NO_3_^−^/L per day. Corroborating with our results, others have observed low or undetectable NO_3_^−^ concentration in milk from treated dairy cows [11,12]. Interestingly, the intermediate treatment (CAN10) presented the highest concentration (0.07 mg/L) of NO_2_^−^ residue in milk. While these findings are difficult to explain, the concentration of 0.07 mg/L is still under the WHO [46] guidelines, which regulates amounts below 3 mg of NO_2_^−^/L to be accepted for human consumption. Overall, our study provided, for the first time, an outline of NO_3_^−^ and NO_2_^−^ residues in milk from treated dairy goats mainly to establish considerations for food safety due to toxicity concerns (e.g., methemoglobinemia in infants), which can be caused by the ingestion of high levels of NO_2_^−^ from food or water [47]. These findings suggest that despite the presence of NO_3_^−^ and NO_2_^−^ residues in the milk, its consumption should not be a concern for food safety; however, as mentioned previously, milk antioxidant capacity may be affected. Indeed, NO_3_^−^ exposure is much greater (~91 mg/person) through the daily ingestion of other foods (e.g., vegetables, water, beer) when compared to the consumption of fresh animal products such as milk, comprising about 7% of the total exposure to NO_3_^−^ residues [48].

To date, there are no previous data evaluating the effects of CAN on milk FA profile from treated dairy goats. Milk FA are generally derived from two main sources: diet and ruminal microbial activity [49]. Changes in the ruminal microbiome might affect milk FA proportions and consequently, the production of milk fat precursors (acetate and butyrate), besides the intermediates of PUFA biohydrogenation in the rumen [50]. However, there is still information needed with regard to the rumen microbial role on milk FA synthesis, especially in goats [51]. The minor response on ruminal fermentation parameters in our study may explain the lack of major effects on milk FA profile. Similar to our results, Klop et al. [16] investigated the effects of dietary NO_3_^−^ to dairy cows and observed minor effects on milk FA composition. In our study, we observed a decreased propionate production associated with greater acetate:propionate ratio, which may partially explain the minimal treatment effects found on the FA profile. Some FAs are largely derived from rumen bacteria such as odd-chain FA (15:0 and 17:0), which may be formed by the elongation of propionate or valerate [52] that decreased and tended to decrease, respectively, in response to CAN supplementation in our study.

Despite the inconsistency of treatment effects on individual FA (tended to decrease 11:0, decreased 12:0, tended to increase 17:0, increased *trans*-9 18:1 and 21:0), no treatment effects were observed on the FA groups (SFA, MUFA, and PUFA). This response validates that CAN (up to 20 g/kg DM) does not seem to cause adverse effects to the FA profile when gradually fed to lactating dairy goats.

### 4.3. Plasma Urea Nitrogen and Ruminal Fermentation Parameters

Supplemental CAN fed at 10 g/kg DM presented the highest concentration of PUN, regardless of the collection time (0 or 4 h after the morning feeding). A suitable explanation for these findings is that providing a lower dose of CAN in the diet (7.65 g/kg NO_3_^−^ on DM basis) led to faster NO_3_^−^ reduction to NH_3_ within the rumen, favoring its absorption through the rumen wall. Conversely, CAN20 seemed to cause a slight reduction in the nitrate-reducing bacteria activity, which likely caused a ruminal NO_2_^−^ accumulation, and consequently limited NH_3_ absorption into the bloodstream. Theoretically, the reduction of NO_2_^−^ to NH_3_ is much slower than the reduction of NO_3_^−^ to NO_2_^−^, which may cause the accumulation of NO_2_^−^ and other intermediates in the rumen, depending on the balance of enzyme activities [3,53].

The concentration of ruminal NH_3_–N was similar regardless of the treatment, which may be considered as a positive effect, assuming no lack of substrate toward the microbial protein synthesis either by feeding CAN or urea. As mentioned previously, the nitrate-ammonia reduction occurs in two steps in the rumen, whereby NO_3_^−^ is converted to NO_2_^−^ rapidly because of higher thermodynamic energy, and subsequently NO_2_^−^ is converted to NH_3_–N [5]. Despite the confounding effect of lower PUN concentration for CAN20, attributed to a slower nitrate-reducing bacteria activity, it seemed that potential toxic effects to the rumen microbiota were alleviated after a while, considering the different time of collection for blood and rumen fluid, performed at 4 h and 8 h postprandial, respectively. In line with our findings, van Zijderveld et al. [54] did not find any effects on NH_3_–N concentration by supplementing 34 g/kg of calcium-ammonium nitrate (~25.5 g/kg of NO_3_^−^ on a DM basis) to growing male lambs.

Calcium-ammonium nitrate supplemented to lactating dairy goats did not affect most ruminal fermentation parameters. As mentioned before, this lack of response is highly related to the absence of secondary effects on nutrient digestibility and milk composition observed in the present study. According to Giger-Reverdin et al. [51], despite the possibility of changes in the rumen environment by switching diets, the ruminal microbiota of goats usually have high stability and resilience, validating the absence of CAN effects on most ruminal fermentation parameters. The reduction in propionate production in our study was also observed by Asanuma et al. [55], along with reductions in acetate and total VFA concentration when potassium nitrate (up to 9 g/day) was fed to male goats. Similar to NO_3_^−^, propionate is also a methane antagonist because of its hydrogen-sink activity in the rumen. Thus, a competition for ruminal [H] was likely the reason for the decreased propionate production, and consequent greater acetate:propionate ratio, even though no changes were observed in acetate proportion.

## 5. Conclusions

Calcium-ammonium nitrate has been extensively used as a methane inhibitor for ruminants; however, further studies to investigate its effects on milk quality from treated animals are still warranted. Our results indicated that supplementing CAN (up to 20 g/kg of DM) to lactating dairy goats did not affect feed intake, nutrient digestibility, yields and composition of milk, and most ruminal fermentation parameters. Low concentration of NO_3_^−^ and NO_2_^−^ residues in milk supports the importance of using safe levels of CAN preceding an adaptation period to prevent negative effects on production performance. Despite the increased CD concentration, indicating sooner lipid oxidation, supplementing CAN (up to 20 g/kg of DM) to dairy goats should not be a concern considering the unchanged TAC in milk and other parameters of milk quality and ruminal fermentation.

## Figures and Tables

**Figure 1 animals-12-00983-f001:**
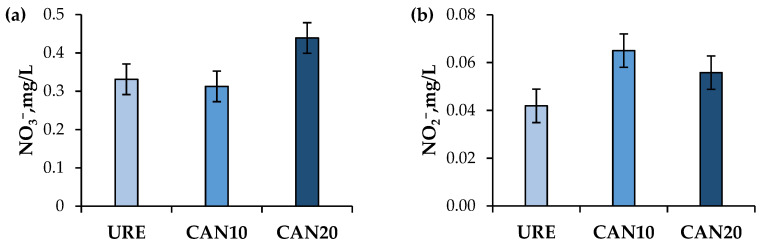
Effects of calcium-ammonium nitrate (CAN) fed to lactating dairy goats on (**a**) concentration of nitrate (NO_3_^−^) in milk (*p*-value: linear = 0.01; quadratic = 0.05; SEM = 0.042) and (**b**) concentration of nitrite (NO_2_^−^) in milk (*p*-value: linear = 0.11; quadratic = 0.04; SEM = 0.007).

**Figure 2 animals-12-00983-f002:**
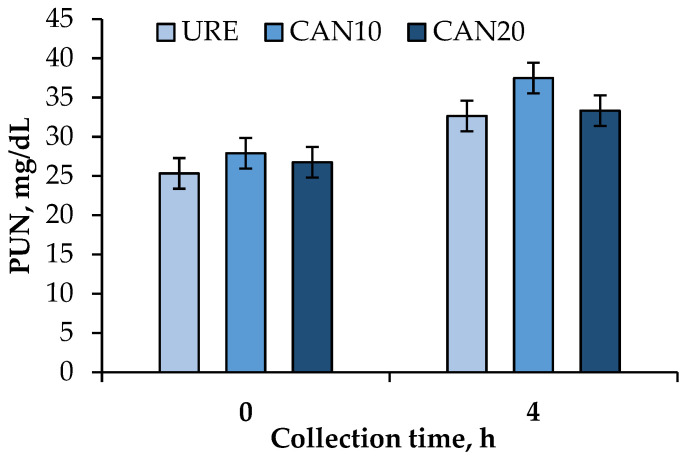
Effects of calcium-ammonium nitrate (CAN) fed to lactating dairy goats on plasma urea nitrogen (PUN) before (0 h) and after (4 h) the morning feeding (*p*-value: treatment = 0.07; time = 0.01; treatment × time = 0.37; SEM = 1.959).

**Table 1 animals-12-00983-t001:** Ingredient proportion and nutritional composition of the experimental diets.

Item	Treatment ^1^		
	URE	CAN10	CAN20
Ingredient proportion, g/kg DM			
Corn silage	450	450	450
Ground corn	382	381	381
Soybean meal	139	139	139
Urea ^2^	7.32	3.66	0.00
Calcium-ammonium nitrate ^3^	0.00	10.0	20.0
Limestone ^4^	11.5	5.77	0.00
Mineral supplement ^5^	10.0	10.0	10.0
Nutritional composition, g/kg DM ^6^			
Dry matter, as-fed basis	505	504	504
Organic matter	944	941	939
Crude protein	160	160	160
Rumen degradable protein	107	107	107
Neutral detergent fiber	299	299	299
Ether extract	32.3	32.3	32.3
Nitrate	0.00	7.65	15.3

^1^ URE = 7.32 g of urea/kg of DM as a control group (without nitrate); CAN10 = 10 g of calcium-ammonium nitrate (CAN)/kg of DM; CAN20 = 20 g of CAN/kg of DM. ^2^ Prote-N, 99.5% DM and 41.7% N on a DM basis (GRASP Ind. & Com. LTDA; Curitiba, Brazil). ^3^ Double salt of calcium-ammonium nitrate decahydrate [5Ca(NO_3_)_2_∙NH_4_NO_3_∙10H_2_O], 85.0% DM; 16.5% N, 19.6% Ca, and 76.5% NO_3_^−^ on a DM basis (Yara; Olso, Norway). ^4^ Composition (per kg of product): 340 g of Ca and 40 g of Mg. ^5^ Composition (per kg of product): 150 g Ca, 60 g P, 50 g S, 5 g Mg, 136 g Na, 90 mg Co, 150 mg Cu, 180 mg I, 400 mg Mn, 13 mg Se, and 3000 mg Zn. ^6^ Unless otherwise stated.

**Table 2 animals-12-00983-t002:** Effects of calcium-ammonium nitrate (CAN) fed to lactating dairy goats on body weight, dry matter intake, and apparent digestibility of nutrients.

Item	Treatment ^1^			SEM	*p*-Value ^2^	
	URE	CAN10	CAN20		Lin	Quad
BW, kg	53.2	53.3	53.0	1.833	0.81	0.73
DMI, kg/d	1.73	1.78	1.76	0.059	0.33	0.21
Digestibility, g/kg DM						
DM	589	618	553	20.93	0.23	0.08
OM	609	635	574	20.27	0.23	0.09
CP	591	649	613	19.89	0.44	0.06
EE	710	728	685	16.21	0.26	0.13
NDF	445	460	417	15.68	0.19	0.11

^1^ URE = control group (without nitrate); CAN10 = 10 g of CAN per kg of DM; CAN20 = 20 g of CAN per kg of DM. ^2^ Lin = linear effect of CAN and Quad = quadratic effect of CAN.

**Table 3 animals-12-00983-t003:** Effects of calcium-ammonium nitrate (CAN) fed to lactating dairy goats on milk production, composition, yield, and antioxidant capacity.

Item	Treatment ^9^			SEM	*p*-Value ^10^	
	URE	CAN10	CAN20		Lin	Quad
Production, kg/d						
Milk yield	2.04	2.14	2.13	0.123	0.16	0.41
3.5% FCM ^1^	1.98	2.06	2.05	0.145	0.29	0.36
ECM ^2^	1.92	2.00	1.99	0.136	0.26	0.40
FE ^3^	1.12	1.14	1.14	0.076	0.52	0.65
Composition, %						
Fat	3.26	3.26	3.19	0.177	0.48	0.68
True protein	2.76	2.73	2.75	0.071	0.92	0.61
Lactose	4.09	4.08	4.07	0.067	0.70	0.98
Yield, kg/d						
Fat	0.067	0.070	0.069	0.006	0.49	0.39
True protein	0.056	0.058	0.059	0.003	0.20	0.66
Lactose	0.084	0.087	0.087	0.005	0.23	0.37
Antioxidant capacity						
TAC ^4^	202	207	200	8.211	0.68	0.14
TBARS ^5^	9.74	7.00	7.34	0.797	0.01	0.06
CD ^6^	47.1	55.7	66.0	4.898	0.01	0.78
MUN ^7^, mg/dL	22.8	22.4	23.3	2.063	0.67	0.55
SCC ^8^, 1000/mL	1570	2172	1417	666.7	0.86	0.37
Log_10_ SCC	2.98	3.00	2.89	0.143	0.63	0.64

^1^ 3.5% Fat-corrected milk [24]; ^2^ Energy-corrected milk [25]; ^3^ Feed efficiency = ECM/DMI; ^4^ Total antioxidant capacity (μM of Trolox equivalent/mL); ^5^ Thiobarbituric acid reactive substances (mmol of malondialdehyde/kg of fat); ^6^ Conjugated dienes (mmol/kg of fat); ^7^ Milk urea nitrogen (mg/dL); ^8^ Somatic cell count (1000/mL); ^9^ URE = control group (without nitrate); CAN10 = 10 g of CAN per kg of DM; CAN20 = 20 g of CAN per kg of DM; ^10^ Lin = linear effect of CAN and Quad = quadratic effect of CAN.

**Table 4 animals-12-00983-t004:** Effects of calcium-ammonium nitrate (CAN) fed to lactating dairy goats on milk fatty acid (FA) profile.

Item ^1^	Treatment ^5^			SEM	*p*-Value ^6^	
	URE	CAN10	CAN20		Lin	Quad
FA proportions						
6:0	0.57	0.50	0.49	0.143	0.69	0.87
8:0	1.41	1.32	1.25	0.222	0.54	0.98
10:0	8.81	8.61	8.03	0.898	0.37	0.80
11:0	0.26	0.22	0.21	0.026	0.08	0.44
12:0	5.63	5.30	4.96	0.431	0.04	0.99
13:0	0.20	0.18	0.18	0.017	0.25	0.48
14:0	14.37	14.28	14.10	0.328	0.34	0.85
14:1	0.88	0.86	0.82	0.061	0.31	0.92
15:0	1.26	1.23	1.36	0.085	0.34	0.40
15:1	0.26	0.27	0.27	0.030	0.88	0.79
16:0	36.8	36.9	38.0	1.255	0.12	0.39
16:1	0.62	0.59	0.66	0.031	0.21	0.07
17:0	0.78	0.81	0.87	0.038	0.09	0.71
17:1	0.15	0.16	0.17	0.021	0.49	0.91
18:0	8.27	9.24	8.58	0.601	0.56	0.08
*trans*-9 18:1	4.70	5.37	5.47	0.237	0.03	0.33
*cis*-9 18:1	12.9	12.2	12.4	0.490	0.37	0.33
*trans*-6 18:2	0.53	0.52	0.54	0.034	0.80	0.74
*cis*-6 18:2	0.94	0.94	0.89	0.069	0.50	0.74
*cis*-9, *trans*-11 CLA	0.11	0.09	0.14	0.055	0.84	0.68
20:0	0.14	0.14	0.14	0.006	0.55	0.97
20:2	0.04	0.05	0.05	0.008	0.15	0.46
21:0	0.17	0.20	0.30	0.048	0.05	0.59
FA groups						
SFA ^2^	76.8	77.1	76.8	0.517	0.92	0.47
MUFA ^3^	19.6	19.5	19.8	0.594	0.75	0.69
PUFA ^4^	1.59	1.56	1.56	0.104	0.78	0.85

^1^ g/100 g of total FA; ^2^ SFA = Saturated fatty acids; ^3^ MUFA = Monounsaturated fatty acids; ^4^ PUFA = Polyunsaturated fatty acids; ^5^ URE = control group (without nitrate); CAN10 = 10 g of CAN per kg of DM; CAN20 = 20 g of CAN per kg of DM; ^6^ Lin = linear effect of CAN and Quad = quadratic effect of CAN.

**Table 5 animals-12-00983-t005:** Effects of calcium-ammonium nitrate (CAN) fed to lactating dairy goats on ruminal pH, NH_3_-N concentration, and VFA profile.

Item	Treatment ^1^			SEM	*p*-Value ^2^	
	URE	CAN10	CAN20		Lin	Quad
pH	6.90	7.17	7.00	0.270	0.54	0.13
NH_3_-N, m*M*	16.0	15.3	14.7	3.772	0.77	0.98
Total VFA, m*M*	49.5	50.5	57.0	6.885	0.34	0.68
Individual VFA, mol/100 mol						
Acetate	62.3	63.1	65.5	1.643	0.14	0.62
Propionate	22.7	21.8	19.2	1.769	0.01	0.39
Isobutyrate	0.82	0.86	0.71	0.128	0.32	0.31
Butyrate	12.2	12.6	12.9	1.000	0.60	0.99
Isovalerate	0.85	0.71	0.71	0.091	0.14	0.33
Valerate	1.07	0.93	0.85	0.113	0.08	0.76
Acetate:Propionate	2.76	2.91	3.61	0.356	<0.01	0.22

^1^ URE = control group (without nitrate); CAN10 = 10 g of CAN per kg of DM; CAN20 = 20 g of CAN per kg of DM; ^2^ Lin = linear effect of CAN and Quad = quadratic effect of CAN.

## Data Availability

The data presented in this study are available on request from the corresponding author.

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
