# Peer review of "Feeding Calcium-Ammonium Nitrate to Lactating Dairy Goats: Milk Quality and Ruminal Fermentation Responses"

_animals, 2022, doi:10.3390/ani12080983_

Round 1

Reviewer 1 Report

The manuscript entitled ‘Feeding calcium-ammonium nitrate to lactating dairy goats: Milk quality and ruminal fermentation responses’ is within the scope of the journal. The topic is maybe not new but the amount of data and analyses is very impressive. Thus, it may increase knowledge about using nitrate additives in small ruminant nutrition. The presentation of data is clear and the discussion of the results is exhaustive. The language of the manuscript is generally satisfying.

I present a few minor suggestions, which should be improved before the manuscript can be published in the journal.

L34: There is lack of units after ‘2.53 ± 0.34’. Please, improve it.

L69, 375, 413: should be ‘mentioned previously’;

Table 3. Please describe the abbreviation ‘MUN’ under the table.

L256-262: In my opinion, authors should describe the results, which are statistically significant firstly and they should just describe the other results. Please improve it throughout the Results.

L321: Please review wording.

L331-332: Please review wording.

L387: ‘suits’? Please review wording.

L423: should be ‘Dietary CAN supplemented to lactating dairy goats (…)’.

Reviewer 2 Report

In this paper entitled " Feeding calcium-ammonium nitrate to lactating dairy goats: Milk quality and ruminal fermentation responses", the Authors study the effects of the administration of calcium ammonium nitrate (CAN) in two different doses (10 and 20 g/kg of Dry Matter, respectively), in the feeding of lactating Saanen goats.

In brief, CAN is a potential methane-inhibitor when used in ruminants feed.

The results did not show significant variations in daily milk yield and its composition, except on the Fatty Acids profile and Antioxidant Capacity of milk.

The Authors also found higher concentrations of nitrate and nitrite in milk, but still below the WHO maximum recommended doses, for food intended for human consumption.

The paper is well structured, interesting, and well written, my minor comments are summarized below:

Line 34: 2.53 ± 0.34 of milk/day, rewrite 2.53 ± 0.34 kg of milk/day.

Line 138-141: “In brief, approximately 0.5 g of feed, feces, and refusals were weighed into Ankom F57 bags (25-µm porosity; Ankom Technology, Macedon, NY, USA) and incubated in the rumen of 2 Holstein ruminally cannulated cows (diet composed by 60% of corn silage and 40% of grain mix on a DM basis) for 288 h”.

I suggest that the Authors rewrite the sentence in a broader and more detailed way, (in particular on the guarantee of animal welfare).

Line 145-146: “Milk samples were collected on days 15 and 16 during each milking (morning and afternoon) and mixed proportionally according to the milk yield”. Add to the sentence, (if known), the characteristics of the milking machine (trademark, vacuum level, pulsator, etc) and the type of milkmeters utilized.

Line 148-150: “A 50 mL aliquot of milk was collected and preserved with 2-bromo-nitropopano-1.3-diol for analyses of fat, protein, and lactose by mid infrared spectrophotometry (Bentley 2000; Bentley Instrument Inc., Chaska, MN, USA).

Add to fat (%), protein (%), and lactose (%), specify the reference material used to calibrate the FTIR instrument for goat's milk; It would have been interesting to know the somatic cells count.

Line 244-245: in table 3, at CD6 “Conjugated dienes”, how would the Authors explain this difference in p-value?

(0.01 vs 0.78 for linear and quadratic effects in treatment).

Line 230: “3.5 %FCM”, rewrite in 3.5 % FCM.

Line 436-444: I suggest to the Authors to insert in the paragraph "Conclusions", a short note on the potential positive environmental implications deriving from the use of CAN, in the ruminants' feed ration.
